# SCPSN: Spectral Clustering-based Pyramid Super-resolution Network for Hyperspectral Images

## ABSTRACT

Single hyperspectral image super-resolution aims to reconstruct a high-resolution hyperspectral image (HRHSI) from an observed low resolution hyperspectral image (LRHSI). Most current methods combine CNN and Transformer structures to directly extract features of all channels in LRHSI for image reconstruction, but they do not consider the interference of redundant information in adjacent bands, resulting in spectral and spatial distortions in the reconstruction results and an increase in model computational complexity. To address this issue, this paper proposes a spectral clustering-based pyramid super-resolution network (SCPSN) to progressively reconstruct HRHSI at different scales. In each image reconstruction layer, a clustering super-resolution block (CSRB) consisting of spectral clustering block (SCB), patch non local attention block (PNAB), and dynamic fusion block (DFB) is designed to achieve the reconstruction of detail features. Specifically, for the high correlation between adjacent spectral bands in LRHSI, a SCB is first constructed to achieve clustering of spectral channels and filtering of hyperchannels. This can reduce the interference of redundant spectral information and the computational complexity of the model. Then, by utilizing the non-local similarity of features within the channel, a patch non-local attention block (PNAB) is constructed to enhance the features of hyperchannels. Next, a dynamic fusion block (DFB) is designed to reconstruct the features of all channels in LRHSI by establishing correlations between enhanced hyperchannels and other channels. Finally, the reconstructed channels are upsampled and added to the corresponding channels to obtain the reconstructed HRHSI. Extensive experiments validate that the performance of SCPSN is superior to that of some other state-of-the-art (SOTA) HSSR methods in terms of visual effects and quantitative metrics. In addition, our model does not require training on large-scale datasets compared to other methods. The dataset and code will be released on GitHub.

## CCS CONCEPTS

• **Computing methodologies** → **Hyperspectral imaging**.

## KEYWORDS

Hyperspectral image super-resolution, Spectral clustering, Non-local feature similarity

**Unpublished working draft. Not for distribution.**

## 1 INTRODUCTION

Image super-resolution (SR) technology [27] aims to improve the spatial resolution and visual quality of low resolution (LR) images. It is a fundamental research area within computer vision, and has been widely applied in various fields, such as remote sensing [4], agricultural monitoring [26], and medical diagnosis [22]. In the field of remote sensing, due to the limitations of the imaging environment of satellite sensors, the captured hyperspectral images (HSIs) usually have rich spectral information but relatively low spatial resolution. Many researchers attempt to use SR technology to reconstruct more spatial texture features in order to improve the spatial resolution of HSI. At present, HSI-SR reconstruction methods are mainly divided into two categories: traditional SR methods and deep learning-based SR methods.

The traditional SR methods mainly include regularization-based methods [11, 33], non-negative matrix factorization-based methods [13] and sparse representation-based methods [8, 32]. These methods rely on manually defined prior information in solving HR images, and improper selection of prior knowledge can lead to serious texture loss in the reconstruction results. In addition, this type of method is time-consuming and difficult to meet the real-time processing requirements in practical applications [7, 23, 36]. In recent years, due to the powerful feature representation ability of convolutional neural networks(CNNs), deep learning-based methods have attracted the attention of researchers. Based on the characteristics of hyperspectral images, some deep networks based on 3D convolution have been proposed. For example, Mei et al. [23] proposed a 3D-FCNN that uses 3D convolution to learn the spatial context of adjacent pixels and the spectral correlation of adjacent bands. Li et al. [14] proposed a hybrid convolutional network (MCNet), which utilizes hybrid 2D/3D convolution to explore more spatial features of HSI. Fu et al. [6]proposed a bidirectional 3D quasi-recurrent neural network with arbitrary number of bands. Although 3D convolution can learn contextual relationships of features in both spatial and channel dimensions, it requires learning a large number of parameters and a significant amount of memory. With the development of deep network structures, the residual structures and attention mechanisms [10, 12, 16, 18] have also been introduced into HSI-SR networks. On this basis, considering the characteristic of HSI, Liu et al. [18] proposed a spectral grouping and attention-driven residual dense network (SGARDN) that uses group convolution to extract spatial features within and between groups composed of highly similar spectral bands, while avoiding spectral confusion caused by normal convolution. CNN based SR methods perform well in extracting local spectral features, but they ignore long-range spatial spectral correlation, resulting in spatial and spectral distortion in reconstructed HR images.

Later, the Transformer structure was proposed and introduced into computer vision tasks, demonstrating good performance in capturing long-distance features. For example, inspired by ViT [5]

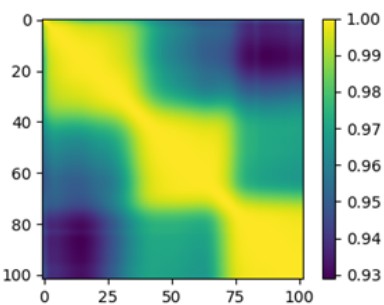

Figure 1: Visualization of spectral correlation coefficient matrix of a HSI

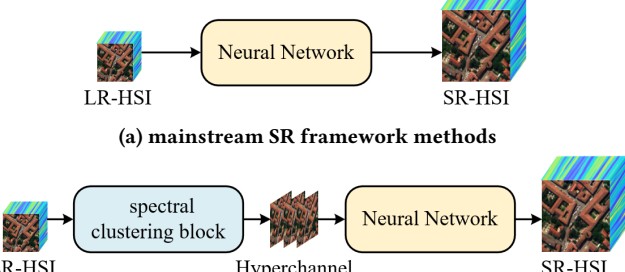

(a) mainstream SR framework methods

(b) SR framework based on hyperchannel filtering

Figure 2: Structure of different SR framework

in advanced visual tasks, Chen et al. [1] introduced Transformer into SR, but it has the problem of high computational complexity. To reduce the computational complexity of the model, Liang et al. [17] introduced SwinTransformer [21] into the SR task, dividing the image into small windows of size 8x8 to participate in the computation of multi-head attention. Transformer based networks need to be trained on large datasets, and the training samples in existing hyperspectral datasets are limited. Therefore, Transformer based HSI-SR methods have only been studied by a small number of researchers. Zhang et al. [34] introduced the spectral correlation coefficient (SCC) of the spectrum to replace the original attention matrix in the Transformer structure, in order to reduce the computational complexity of the model. However, this method does not fully utilize spatial information. Liu et al. [20] employs a combination of 3-D convolutions and transformer block to extract complementary spatial and spectral features. Although the Transformer structure is effective in exploring long-range feature dependencies in spatial dimensions, it is not conducive to local feature extraction. Additionally, it has high hardware memory requirements and computational complexity.

At present, most deep learning based HSI-SR methods improve the model's feature extraction and representation capabilities by improving the network structure, lacking consideration for the spectral characteristics of HSI. Due to the high spectral correlation between adjacent channels in HSI, as shown in Figure 1, the interference of redundant information can cause spectral distortion in the reconstructed image. Therefore, this paper constructs an SR framework based on hyperchannel filtering, as shown in Figure 2(b). In addition, each band in HSI reflects spectral information of similar substances, and the features within the channel have non local similarity characteristics. The non local self-similarity of features has been proven to be an effective prior for image restoration [3, 9]. This prior has been successfully introduced into network construction [31] and has been applied in many image restoration tasks [19, 24, 35]. Based on the above analysis, we proposed a spectral clustering-based pyramid super-resolution network (SCPSN), which gradually achieves image reconstruction by constructing image reconstruction layers at different scales. In each image reconstruction layers, a clustering super-resolution block (CSRB) and a residual block are constructed to generate the reconstructed HR image of the current layer. CSRB consists of spectral clustering

block (SCB), patch non local attention block (PNAB), and dynamic fusion block (DFB), and is designed to achieve the reconstruction of detail features. In CSRB, SCB is constructed to achieve clustering of spectral channels and filtering of hyperchannels, in order to reduce the interference of redundant spectral information and the computational complexity of the model. PNAB is proposed to reconstruct hyperchannels with more detail features by utilizing the non-local similarity of features within the channel. DFB is designed to integrate each channel in LRHSI with highly correlated channels in hyperchannels to reconstruct all channels of LRHSI. Qualitative and quantitative experiments conducted on three hyperspectral datasets have demonstrated that our method outperforms some state-of-the-art (SOAT) methods. The main contributions of this paper are as follows.

- We propose a SCPSN that includes multiply image reconstruction layers to progressively reconstruct HRHSI with rich textures at different scales.
- Based on high correlation between adjacent channels, an SCB is constructed to select hyperchannels with rich information from all channels of LRHSI, which can effectively reduce the computational complexity and number of parameters of the model.
- Considering the non-local similarity of features within the channel, a PNAB is constructed to enhance the features of hyperchannels by learning feature correlations between image patches.
- A DFB is designed to reconstruct the detail features for each spectral band by a channel dynamic filtering block and a feature fusion block, which dynamically select highly correlated hyperchannels and fuse them with each spectral band in LRHSI.

## 2 PROPOSED METHOD

In this section, to reconstruct more detail features of LRHS images, we proposed a SCPSN, as shown in Figure 3, which adopts a pyramid structure to gradually reconstruct images of different scales. Each scale layer of this structure contains a CSRB and a residual block (RB) to achieve the reconstruction of HR image at the current scale. CSRB consisting of a SCB, a PNAB, and a DFB is designed to achieve the reconstruction of detail features. RB is used to integrate the reconstructed features, and these features are combined with the

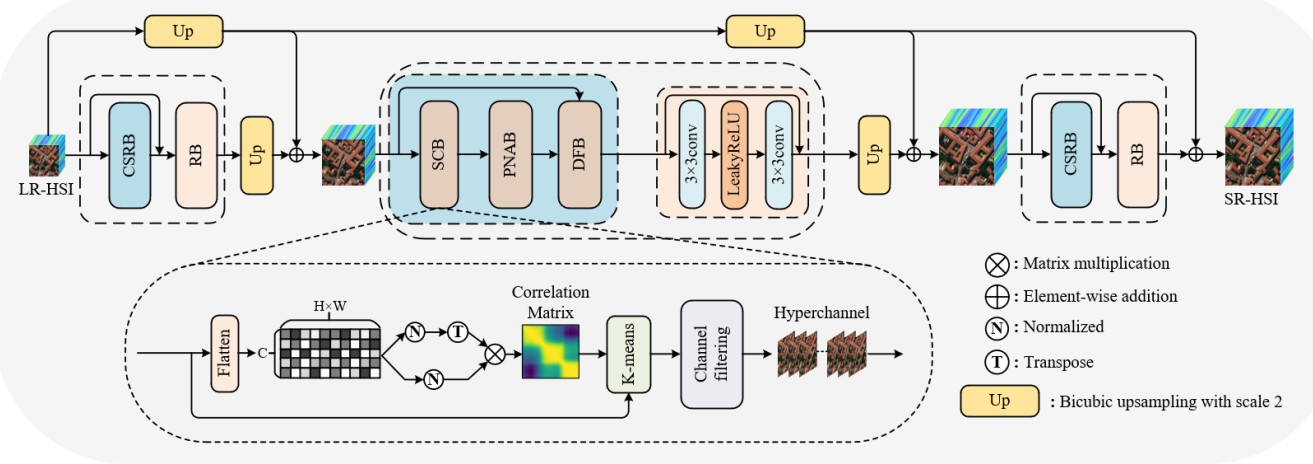

**Figure 3: Overview of the proposed SCPSN at SR×4**

input LR image to obtain the reconstructed HR image through a residual structure. Below, we provide a detailed introduction to the specific structures of SCB, NAB and DFB.

## 2.1 Spectral Clustering Block(SCB)

Hyperspectral images typically consist of hundreds of spectral bands, providing rich spectral information, but with relatively low spatial resolution. However, due to the strong correlation between adjacent spectral bands, there is a large amount of redundant information. Directly processing all spectral bands in the SR task will increase the computational complexity of the model. In addition, due to the influence of imaging environment, there is a large amount of invalid information in some bands, which can also interfere with feature reconstruction. Therefore, we propose a SCB, as shown in Figure 3, which clusters the bands of HSI by calculating the correlation between adjacent bands, and selects spectral hyperchannels with rich information to provide detail features for future feature reconstruction. The selection of spectral hyperchannels greatly reduces the number of spectral bands involved in calculations, thus reducing the computational complexity of the model. The structure of SCB is described as follows.

Given an input HSI $I_{LR} \in \mathbb{R}^{H \times W \times C}$. First, $I_{LR}$ is flattened into a matrix $F_f \in \mathbb{R}^{HW \times C}$. Then, the matrix $F_f$ is normalized, and the Pearson correlation coefficients of $F_f$ are calculated through product operation to generate the spectral correlation matrix $M \in \mathbb{R}^{C \times C}$ where the $i$-th row of $M$ represents the similarity between the $i$-th spectral band and other spectral bands in the input image of SCB. $M$ can be obtained by the following equation

$$M = \frac{\left(F_f - \bar{F}_f\right)\left(F_f - \bar{F}_f\right)^T}{\left\|\left(F_f - \bar{F}_f\right)\right\| \cdot \left\|\left(F_f - \bar{F}_f\right)\right\|} \tag{1}$$

where $\bar{F}_f$ represents the mean of $F_f$, $T$ represents the transpose operation of the matrix, and $\|\cdot\|$ represents the L1 norm.

Subsequently, the K-means algorithm is adopted to divide the spectral channels into $K$ clusters based on the spectral correlation

matrix. $K$ is a hyperparameter manually set based on experience. Finally, for each cluster, the average Euclidean distance (AED) between each band and other bands is calculated, indicating its similarity to all bands in the cluster. The band with the minimum AED represents the spectral hyperchannel of the cluster, so that $K$ hyperchannels are filtered out from $K$ clusters , named $I_{SCB}^k$, which are used to reconstruct detail features. The above process can be defined as the following equation

$$I_{SCB}^k = \text{Min}_k\left(Ed_k\left(KM\left(M, I_{LR}\right)\right)\right), k = 1, \dots, K \tag{2}$$

where $KM(\cdot)$ represents the K-means algorithm, $Ed_k(\cdot)$ represents the calculation of AED between each band and other bands in the $k$-th cluster, and $Min_k(\cdot)$ represents the operation of obtaining the minimum value of AEDs in the $k$-th cluster.

## 2.2 Patch Non-local Attention Block(PNAB)

Although the obtained $K$ hyperchannels have richer information compared to other channels, their spatial resolution is still relatively low. Each channel in HSI reflects spectral information of a narrow spectral band, and there are similar texture structures at different positions in its spatial dimension. Therefore, the spatial features in each channel have the characteristic of non-local similarity. The non-local similarity of images has been proven to be an effective prior in the field of image restoration. The self-attention mechanism in Transformer and early non-local operation can capture long-distance dependencies of data through global operations, but the calculation of data autocorrelation in the self-attention mechanism can reduce the computational complexity of the algorithm compared to the calculation of Euclidean distance in non-local operations. Therefore, this paper constructs a patch non-local attention block (PNAB) based on the transformer architecture to enhance the features in $K$ channels, as shown in Figure 4. The specific operation is as follows.

Firstly, the cutting and unfolding operations are performed on the K hyperchannels. Specifically, the K hyperchannels are cut into two types of $7 \times 7$ patches using stride sizes of 4 and 1, which are represented as $H(\cdot)$ and $G(\cdot)$ . The cutting blocks are unfolded

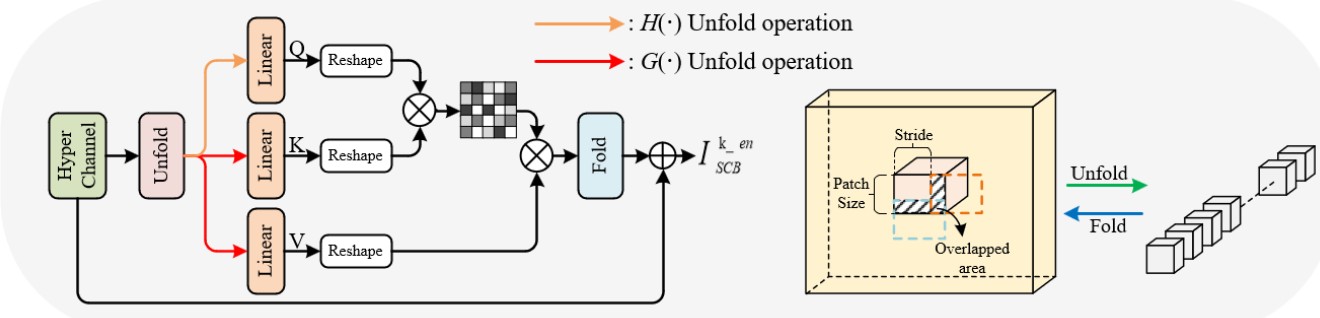

**Figure 4: The structure of PNAB.**

and arranged into two block sequences. Here, using different stride sizes can capture richer boundary features while reducing blocking artifacts.

Then, based on the idea of transformer, we construct a non-local attention block by introducing self-attention of patches to learn feature correlations between patches. And these patches need to be transformed into three token sequences through the linear mapping layers. The operation of non-local attention block can be defined as:

$$
F_{en} = \text{Softmax}\left(\frac{FC\left(UFd\left(H\left(I_{SCB}^k\right)\right)\right) \cdot FC\left(UFd\left(G\left(I_{SCB}^k\right)\right)\right)^T}{\sqrt{d_k}}\right) \cdot
$$
$$
FC\left(UFd\left(G\left(I_{SCB}^k\right)\right)\right)\frac{1}{z} \tag{3}
$$

where $UFd\left(\cdot\right)$ and $FC\left(\cdot\right)$ represent the unfolding operation and linear mapping layer (i.e., fully connected layer) respectively. $d_k$ is the dimension of the input vector and $Softmax\left(\cdot\right)$ is normalized exponential function. $z$ represents the normalization constant calculated by $z = \phi\left(I_{SCB}^k\right)$, $\phi\left(\cdot\right)$ stands for standard deviation calculation process. $F_{en}$ represents the enhanced feature patches.

Finally, the enhanced feature blocks are integrated into feature maps of size $H \times W$ through a folding operation which is the inverse process of the unfolding operation with a stride size of 4 , and the overlapping areas are processed by directly taking the average value. These feature maps are added with the $K$ hyperchannels through a residual operation to achieve feature enhancement of hyperchannels. This operation can be represented by the following equation

$$
I_{SCB}^{k\_en} = I_{SCB}^k + Fd\left(F_{en}\right), k = 1, \ldots, K \tag{4}
$$

where $Fd\left(\cdot\right)$ represents the folding operation, and $I_{SCB}^{k\_en}$ represents the enhanced hyperchannels.

## 2.3 Dynamic Fusion Block(DFB)

To reconstruct detail features of all channels at the current scale, DFB consisting of a channel dynamic filtering block and a feature fusion block is designed by establishing correlations between enhanced hyperchannels and other channels. The specific execution process of DFB is described below and shown in figure 5.

Firstly, the channel dynamic filtering block is constructed to dynamically select channels with higher correlation with each channel $I_{LR}^i$ in LRHSI from $K$ hyperchannels for later feature reconstruction. Specifically, the values in the hyperchannels and LRHSI are normalized and expanded to obtain the corresponding matrices $I_{SCB}^{k\_en} \in \mathbb{R}^{HW \times K}$ and $I_{LR} \in \mathbb{R}^{C \times HW}$, and the point multiplication is performed on them to obtain the correlation coefficient matrix $M_C \in \mathbb{R}^{C \times K}$. $M_C^i$ represents the correlation coefficients between the $i$-th channel in LRHSI and the $K$-th hyperchannel. According to the experiment, we found that the number of values with high correlation coefficients in $M_C^i$ is usually less than $K/2$. Therefore, for the convenience of calculation, the top $K/2$ channels with the highest numerical ranking in $M_C^i$ are dynamically selected and sent together with $I_{LR}^i$ into the feature fusion block to reconstruct the $i$-th channel in LR-HSI. The above operation can be represented by the following equation

$$
M_c = \text{Flatten}\left(I_{SCB}^{k\_en}\right) \cdot \text{Flatten}\left(I_{LR}\right)^T \tag{5}
$$

$$
I_c^i = \text{Filtrate}\left(M_c, I_{SCB}^{k\_en}\right), i = 1, \ldots, C \tag{6}
$$

$$
I_G^i = \text{Concate}\left(I_{LR}^i, I_C^i\right), i = 1, \ldots, C \tag{7}
$$

where Filtrate represents the selection operation that includes sorting of correlation coefficients and dynamic channel filtering, and $I_C^i$ represents the selected hyperchannels with high correlation with the $i$-th channel $I_{LR}^i$ in LRHSI. and $I_G^i$ represents the $i$-th group of channels containing $I_{LR}^i$ and $I_C^i$.

Then, a channel reconstruction block is constructed to reconstruct each channel of the LR image. Specifically, each group of channels ($I_G^i$, $i = 1, \ldots, C$) are fused together through the convolutional layers and a LeakyRelu activation function to obtain $F_{re}^i \in \mathbb{R}^{H \times W \times 1}$. In addition, the hyperchannels in each group are used to enhance the fused features through a Maxpooling operation and a Sigmoid function. The process of channel reconstruction block can be represented by the following equation

$$
F_{re}^i = \text{Conv}_{3\times3}\left(\text{ReLu}\left(\text{Conv}_{3\times3}\left(I_G^i\right)\right)\right), i = 1, \ldots, C \tag{8}
$$

$$
C_{SR}^i = F_{re}^i \cdot \text{Sig}\left(Mp\left(I_C^i\right)\right) + F_{re}^i, i = 1, \ldots, C \tag{9}
$$

where $C_{SR}^i$ represents the reconstructed $i$-th channel.

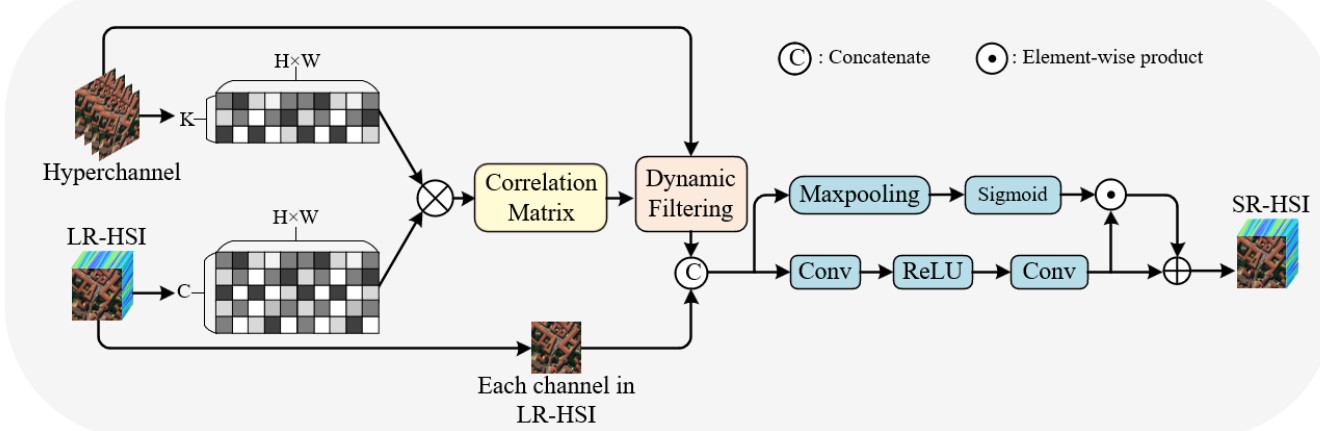

**Figure 5: The structure of DFB. Dynamic Filtering represents the operation of dynamically selecting the most relevant k/2 Hyperchannels.**

Finally, all reconstructed channels pass through a RB, a upsampling operation and a residual structure to obtain the reconstructed HR image at the current scale, which can be represented as:

$$I_{SR}^{S_i} = Up\left(RB\left(F_{re}^i\right)\right) + Up\left(I_{LR}^i\right), i = 1, \ldots, C \quad (10)$$

$I_{SR}^{S_i}$ denotes the reconstructed HR image at the $s\text{-}th$ (s=2,4,8) scale.

## 2.4 Loss function

To generate reconstructed HR images with rich spatial information and spectral fidelity, a joint loss function is defined to train SCPSN. This function consists of four parts: reconstruction loss $\mathcal{L}_{rec}$, spectral fidelity loss $\mathcal{L}_{spe}$, and multi-scale reconstruction loss $\mathcal{L}_{rec}^s$, which are defined as follows:

$$\mathcal{L}_{\text{total}} = \mathcal{L}_{rec} + \alpha\mathcal{L}_{\text{SAM}} + \beta\mathcal{L}_{rec}^s \quad (11)$$

Where $\alpha = 1/D\left(\mathcal{L}_{\text{SAM}}/\mathcal{L}_{rec}\right)$ and $\beta = 1/D\left(\mathcal{L}_{rec}^s/\mathcal{L}_{rec}\right)$ are adaptive hyperparameters, and $D(\cdot)$ means gradient calculation that is not required for back propagation.

Extensive work has demonstrated the positive role of $l_1$ and $l_2$ losses in SR tasks [32]. However, L1 loss has been proven to perform better as a reconstruction loss than L2 loss. Therefore, L1 loss is used to calculate the pixel-level difference between the reconstructed HRHSI $I_{SR}$ and the corresponding ground-truth (GT) image $I_{GT}$, which can be expressed as:

$$\mathcal{L}_{rec} = \frac{1}{N}\sum_{n=1}^{N}\left\|I_{\text{GT}}^n - I_{\text{SR}}^n\right\|_1 \quad (12)$$

N is the number of images in a training batch, and $\|\cdot\|_1$ denotes the L1 norm.

To ensure spectral fidelity of the reconstructed HRHSI, a SAM loss is defined to measure the spectral similarity between $I_{SR}$ and $I_{GT}$, and the smaller its value, the more similar the spectra of the two images are. SAM loss can be expressed as:

$$\mathcal{L}_{\text{SAM}}(\Theta) = \frac{1}{N}\sum_{n=1}^{N}\frac{1}{\pi}\arccos\left(\frac{I_{\text{GT}}^n \cdot I_{\text{SR}}^n}{\left\|I_{\text{GT}}^n\right\|_2 \cdot \left\|I_{\text{SR}}^n\right\|_2}\right) \quad (13)$$

where $arccos(\cdot)$ denotes the inverse cosine function and $\|\cdot\|_2$ denotes the L2 norm.

To quickly converge the network and ensure the generation of high-quality reconstructed images at each scale, the multi-scale reconstruction loss is defined and represented as:

$$\mathcal{L}_{rec}^S = \sum_{n=1}^{\log_2 s}\mathcal{L}_{rec}^n \quad (14)$$

where $s\,(2, 4, or\, 8)$ denotes the scale factor.

# 3 EXPERIMENTS AND RESULTS

## 3.1 Dataset and Experimental Setup

In this section, to verify the performance of the proposed SCPSN, we conducted a large number of experiments on three HSI synthetic datasets including Pavia Center [25], Botswana [29] and the University of Pavia [25]. All datasets are processed following the Wald protocol [30], and HRHSIs are blurred using a Gaussian blur kernel of size 8×8 and downsampled to generate the corresponding LRHSIs.

Pavia University dataset: The Pavia University dataset is the raw data of this hyperspectral image obtained by sampling the University of Pavia and its surrounding areas in Italy using the Reflection Optical System Imaging Spectroradiometer (ROSIS). The sampling range of this spectral imager is 430 ~ 860 nm, with a total of 102 spectral segments. The spatial resolution is 1.3 m and the image size is 610×340 pixels. We divide the hyperspectral image into 45 non-overlapping cubic blocks of size 64×64×102 to obtain the reference images (i.e., GTs) in Pavia University dataset. 33 cube blocks are randomly selected for training, while the remaining 12 blocks are used for testing.

**Table 1: Quantitative comparison of different methods on Pavia University and Botswana datasets. The best results are highlighted in bold.**

| Method | | Pavia University | | | | | | Botswana | | | | | |
|---|---|---|---|---|---|---|---|---|---|---|---|---|---|
| | | SSIM↑ | CC↑ | SAM↓ | RMSE↓ | ERGAS↓ | PSNR↑ | SSIM↑ | CC↑ | SAM↓ | RMSE↓ | ERGAS↓ | PSNR↑ |
| **Bicubic** | | 0.8876 | 0.9457 | 4.0108 | 0.0273 | 4.0879 | 31.8355 | 0.9005 | 0.9075 | 1.8687 | 0.019 | 2.1869 | 40.0174 |
| **DHP** | | 0.9331 | 0.9641 | 3.9832 | 0.0206 | 3.2418 | 34.1417 | 0.9333 | 0.9339 | 1.7135 | 0.0149 | 1.8493 | 41.7317 |
| **MCNet** | | 0.9348 | 0.9641 | 3.5852 | 0.021 | 3.2026 | 34.1387 | 0.9305 | 0.9364 | 1.584 | 0.0154 | 1.7622 | 41.8825 |
| **ERCSR** | ×2 | 0.9459 | 0.9706 | 3.3074 | 0.019 | 2.8932 | 35.0298 | 0.93 | 0.9255 | 1.6867 | 0.0166 | 1.9199 | 41.1373 |
| **MSDformer** | | 0.9492 | 0.9722 | 3.3072 | 0.0182 | 2.8404 | 35.2939 | 0.9348 | 0.9368 | 1.6412 | 0.0148 | 1.8582 | 41.7145 |
| **ESSA** | | 0.9493 | 0.9728 | 3.3788 | 0.018 | 2.8009 | 35.4185 | 0.9353 | 0.9374 | 1.6571 | 0.0147 | 1.7788 | 41.93 |
| **Ours** | | **0.9534** | **0.9751** | **3.0463** | **0.0172** | **2.6857** | **35.7657** | **0.9371** | **0.9444** | **1.499** | **0.0143** | **1.5826** | **42.5988** |
| **Bicubic** | | 0.6827 | 0.8247 | 6.1923 | 0.0499 | 7.1626 | 26.6125 | 0.7846 | 0.7496 | 2.8237 | 0.0301 | 3.69 | 35.8657 |
| **DHP** | | 0.8001 | 0.8988 | 5.818 | 0.0377 | 5.5415 | 28.9206 | 0.8196 | 0.8033 | 2.7133 | 0.026 | 3.5472 | 36.5655 |
| **MCNet** | | 0.793 | 0.9014 | 5.088 | 0.0369 | 5.3129 | 29.235 | 0.8211 | 0.7992 | 2.595 | 0.027 | 3.5115 | 36.4421 |
| **ERCSR** | ×4 | 0.7931 | 0.9032 | 5.2234 | 0.0366 | 5.2576 | 29.3096 | 0.8111 | 0.8142 | 2.6505 | 0.0268 | 3.2308 | 36.8997 |
| **MSDformer** | | 0.8119 | 0.9099 | 5.0071 | 0.0353 | 5.1488 | 29.5453 | 0.8206 | 0.7917 | 2.672 | 0.0271 | 3.394 | 36.5454 |
| **ESSA** | | 0.8314 | 0.9167 | 4.7383 | 0.0341 | 4.9621 | 29.879 | 0.8281 | 0.8366 | 2.4944 | 0.0248 | 3.0039 | 37.5031 |
| Ours | | **0.8471** | **0.9266** | **4.2945** | **0.0317** | **4.6213** | **30.5044** | **0.8306** | **0.8449** | **2.3189** | **0.0243** | **2.8759** | **37.798** |
| **Bicubic** | | 0.5133 | 0.585 | 9.5789 | 0.0708 | 10.2254 | 23.6209 | 0.7193 | 0.5525 | 3.5515 | 0.0386 | 4.8633 | 33.6759 |
| **DHP** | | 0.5864 | 0.7029 | 8.6927 | 0.0618 | 9.0307 | 24.7284 | 0.724 | 0.6426 | 3.4746 | 0.0357 | 4.8313 | 34.1217 |
| **MCNet** | | 0.585 | 0.7183 | 8.2405 | 0.06 | 8.7215 | 25.0765 | 0.7291 | 0.6153 | 3.4378 | 0.0366 | 4.8036 | 33.8576 |
| **ERCSR** | ×8 | 0.5695 | 0.7047 | 8.3029 | 0.061 | 8.8925 | 24.9007 | 0.7148 | 0.5772 | 3.4962 | 0.0377 | 4.8226 | 33.8045 |
| **MSDformer** | | 0.5846 | 0.7178 | 8.3847 | 0.0597 | 8.7354 | 25.0648 | 0.7383 | 0.651 | 3.4251 | 0.0354 | 4.6757 | 34.1404 |
| **ESSA** | | 0.5902 | 0.7222 | 8.368 | 0.0596 | 8.6734 | 25.0831 | 0.7389 | 0.6392 | 3.4387 | 0.0355 | 4.5252 | 34.2975 |
| **Ours** | | **0.6122** | **0.7365** | **7.618** | **0.0577** | **8.4441** | **25.3352** | **0.7421** | **0.7073** | **3.112** | **0.0334** | **4.131** | **35.135** |

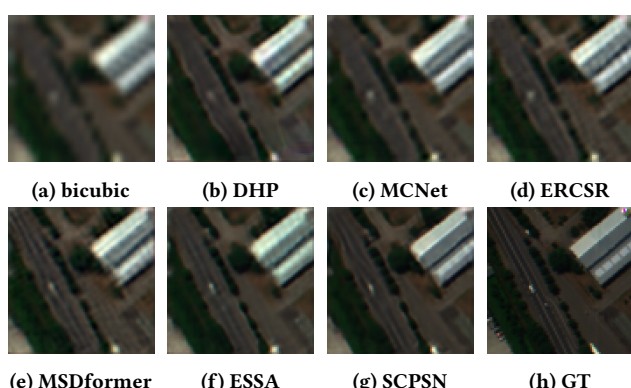

| (a) bicubic | (b) DHP | (c) MCNet | (d) ERCSR |
|---|---|---|---|
| (e) MSDformer | (f) ESSA | (g) SCPSN | (h) GT |

**Figure 6: Visual comparison of SR x4 of an image from the Pavia University dataset**

Botswana dataset: Botswana scenes were acquired by the Hyperion sensor on NASA's Earth Observation 1 (EO-1) satellite. The original Botswana HSI consists of 242 spectral bands ranging from 400 to 2500 nm with a spectral resolution of 10 nm. The spatial size of the original Botswana image is 1496×256 pixels. We removed uncalibrated and noisy bands from the original image to obtain an HSI with 145 bands, and it is further divided into 92 non-overlapping cubic blocks of size 64×64×145 to obtain the reference images (i.e., GTs) in Botswana dataset. 69 cube blocks are randomly selected for training, while the remaining 23 blocks are used for testing.

Pavia Centre dataset: The Pavia Centre Dataset is a hyperspectral dataset acquired by the ROSIS sensor during a flight campaign over Pavia in northern Italy. Thirteen noisy bands were discarded from the original HSI, resulting in an HSI with 102 bands from 430 nm to 860 nm. In addition, the rectangular area of 1096 × 381 pixels with no information in the center of the original HSI was also discarded, and the resulting "two-part" image of size 1096 × 715 × 102 was used for experiments, and divided into 187 non-overlapping cubic blocks of size 64×64×102 to obtain the reference images (i.e., GT) in Pavia Center dataset. 138 cube blocks are randomly selected for training, while the remaining 49 blocks are used for testing.

To display the experimental results for subjective comparison, three bands from HSIs are selected to form RGB images. The 10th, 30th and 60th bands are selected as RGB channels for the Pavia University dataset, the 10th, 35th and 61th bands are selected as RGB channels for the Botswana dataset, and the 10th, 30th and 60th bands are selected as RGB channels for the Pavia Centre dataset. For a fair comparison, all DL-based methods were retrained using Python 3.9.13 and PyTorch 1.13.1 On Ubuntu 20.04 system with NVIDIA GeForce GTX A6000. The proposed SCPSN is trained for 1000 epochs using the Adam optimizer. The initial learning rate is set to 0.001, and inference is performed every five rounds of training. When the loss of the verification set no longer decreases for five consecutive rounds, the learning rate is attenuated by half.

**Table 2: Quantitative comparison of different methods on the Pavia Centre dataset. The best results are highlighted in bold.**

| Method | | Pavia Centre | | | | | |
|---|---|---|---|---|---|---|---|
| | | SSIM↑ | CC↑ | SAM↓ | RMSE↓ | ERGAS↓ | PSNR↑ |
| **Bicubic** | | 0.9010 | 0.9446 | 4.7695 | 0.0240 | 4.6193 | 34.0790 |
| **DHP** | | 0.9509 | 0.9605 | 4.3986 | 0.0164 | 3.5156 | 36.9808 |
| **MCNet** | | 0.9454 | 0.9619 | 4.2348 | 0.0179 | 3.5927 | 36.5016 |
| **ERCSR** | ×2 | 0.9516 | 0.9661 | 4.1091 | 0.0168 | 3.3461 | 37.1148 |
| **MSDformer** | | 0.9541 | 0.9633 | 4.2080 | 0.0162 | 3.3908 | 37.2392 |
| **ESSA** | | 0.9555 | 0.9646 | 4.1269 | 0.0156 | 3.3007 | 37.5543 |
| **Ours** | | **0.9578** | **0.9679** | **3.9015** | **0.0153** | **3.1983** | **37.7238** |
| **Bicubic** | | 0.6987 | 0.8194 | 7.0668 | 0.0450 | 8.2805 | 28.8448 |
| **DHP** | | 0.8149 | 0.8831 | 6.4407 | 0.0348 | 6.6482 | 30.6850 |
| **MCNet** | | 0.7946 | 0.8813 | 6.4178 | 0.0354 | 6.6225 | 30.8185 |
| **ERCSR** | ×4 | 0.8008 | 0.8836 | 6.4497 | 0.0348 | 6.5168 | 30.9526 |
| **MSDformer** | | 0.8046 | 0.8827 | 6.4899 | 0.0347 | 6.5763 | 30.8406 |
| **ESSA** | | 0.8307 | 0.8958 | 6.0923 | 0.0325 | 6.1593 | 31.4355 |
| **Ours** | | **0.8339** | **0.8989** | **5.6875** | **0.0322** | **6.0688** | **31.5756** |
| **Bicubic** | | 0.5369 | 0.6008 | 9.5930 | 0.0643 | 11.7798 | 25.8229 |
| **DHP** | | 0.6178 | 0.7097 | 9.5091 | 0.0548 | 10.9616 | 26.6353 |
| **MCNet** | | 0.6078 | 0.7246 | 8.8605 | 0.0541 | 10.0555 | 27.1181 |
| **ERCSR** | ×8 | 0.5846 | 0.6979 | 8.9714 | 0.0560 | 10.5264 | 26.7494 |
| **MSDformer** | | 0.6097 | 0.7145 | 8.8060 | 0.0547 | 10.2657 | 26.9689 |
| **ESSA** | | 0.6278 | 0.7269 | 8.5905 | 0.0538 | 10.0458 | 27.0852 |
| **Ours** | | **0.6359** | **0.7396** | **8.2716** | **0.0528** | **9.8175** | **27.3280** |

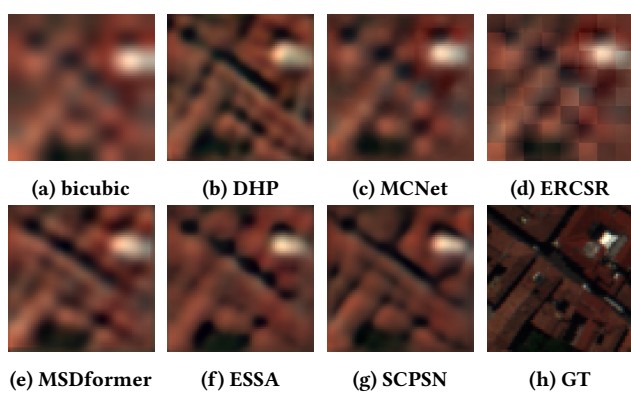

(a) bicubic  (b) DHP  (c) MCNet  (d) ERCSR

(e) MSDformer  (f) ESSA  (g) SCPSN  (h) GT

**Figure 7: Visual comparison of SR x8 of an image from the Pavia Center dataset**

It is worth emphasizing that our network training takes very little time, only about 2 hours.

## 3.2 Comparison with state-of-the-art methods

To evaluate the efficacy of the proposed SCPSN, we subjectively and objectively compare it with several state-of-the-art (SOTA) methods, including DHP [28], MCNet [14], ERCSR [15],MSDformer [2], and ESSA [34]. The metrics used for objective evaluation include correlation coefficient (CC), spectral angle mapping (SAM), structural similarity (SSIM), root mean square error (RMSE), erreur relative globale adimensionnelle de synthese (ERGAS), and peak Signal-to-noise ratio (PSNR), which has been widely used in HSSR tasks to evaluate the quality of spectral and spatial information. The symbol ↑ indicates that the higher the value, the better the

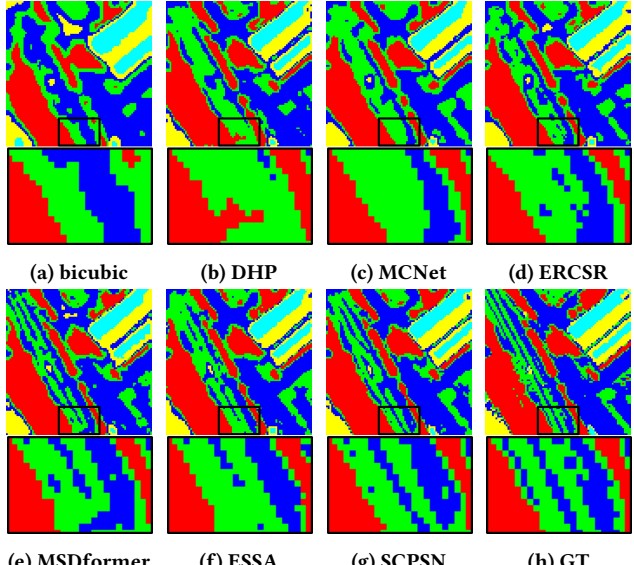

(a) bicubic  (b) DHP  (c) MCNet  (d) ERCSR

(e) MSDformer  (f) ESSA  (g) SCPSN  (h) GT

**Figure 8: Classification experiments on Pavia university dataset.**

performance of the method, while ↓ indicates that the lower the value, the better the results. All comparative experiments are performed on three commonly used datasets. The experimental results are shown in Table 1 and Table 2. From the table, it can be seen that the proposed SCPSN outperforms other methods in all metrics for the three scale factors, which also proves the effectiveness of our model. Compared with the suboptimal results on three datasets, our method achieves an increase of approximately 0.25dB, 0.84dB, and 0.21dB in PSNR values under an scale factor of 8. Especially, ERGAS can objectively reflect the spatial quality of HSIs, while SAM can measure the degree of spectral distortion. Compared with other methods, the proposed network achieved better ERGAS and SAM values. Therefore, the SCPSN can generate better HRHSI images with spectral and spatial fidelity.

Figure 6, shows the visualization results of SR×4 obtained by different methods on Pavia University dataset. From the figure, it can be clearly observed that our results have clearer edges and colors closer to the GT image, indicating that our method reconstructs more spatial details and spectral information. Figure 7 shows the visualization results of SR×8 obtained by different methods on Pavia Centre dataset. Even with a relatively large amplification factor, our method can still reconstruct clear edges and spectral information closer to the GT image. This further demonstrates that the proposed SCPSN has better performance.

## 3.3 Ablation study

To verify the effectiveness of the proposed network, we conducted several ablation experiments on the main components of the network, such as SCB, the number of clusters in SCB, DFB, hyperparameter settings, and the computational complexity of the model. All experiments are conducted on the Pavia Center dataset with a scale factor of ×4.

**Table 3: Ablation study for SCB.**

| Method | SAM↓ | PSNR↑ | Params(M) | FLOPs(G) |
|--------|------|-------|-----------|----------|
| W/O SCB | 5.6972 | 31.5584 | 40.14 | 41.09 |
| W/ SCB | **5.6875** | **31.5756** | **3.81** | **6.37** |

**Table 4: The impact of the number of clusters in SCB.**

| Number | SAM↓ | PSNR↑ | Params(M) | FLOPs(G) |
|--------|------|-------|-----------|----------|
| 4 | 5.7440 | 31.4503 | 2.90 | 5.47 |
| 8 | 5.7084 | 31.5366 | 3.09 | 5.66 |
| 16 | **5.6875** | **31.5756** | 3.81 | 6.37 |
| 32 | 5.7162 | 31.4422 | 6.66 | 9.13 |

**Table 5: Ablation study for DFB.**

| Method | SAM↓ | ERGAS↓ | PSNR↑ |
|--------|------|--------|-------|
| W/O DFB | 5.7743 | 6.1359 | 31.4664 |
| W/ DFB | **5.6875** | **6.0688** | **31.5756** |

Ablation study on the effectiveness of SCB. We conducted experiments on CSRB with and without SCB in each layer of SCPSN. As shown in Table 3, the experimental results indicate that SCB significantly reduces the number of parameters and computational complexity of the model, but does not reduce the performance of the network.

Ablation study on the impact of the number of clusters in SCB on model performance. We conducted experiments on the number of clusters in SCB, selecting 4, 8, 16, and 32, respectively. The experimental results are shown in Table 4. According to the experiments, the network performance is optimal when the number of clusters is 16.

Ablation study on the effectiveness of DFB. An ablation experiment was designed for DFB by replacing it with a convolutional block containing a 3×3 convolution, ReLu function, and a 3×3 convolution. The experimental results are shown in Table 5. As can be seen from the table, the indicators of the models using DFB have been significantly improved.

Ablation study on the hyperparamrters. Table 6 presents the results obtained by manually and Adaptively setting hyperparameters in the loss function. We manually set the hyperparameters $\alpha$ and $\beta$ to 0.001. The results obtained by directly using the defined adaptive hyperparameter calculation method are better than the results obtained by manually adjusting the hyperparameters. This also indicates that the hyperparameter calculation method in this paper is effective.

Ablation study on the computational complexity of the model. Table 7 lists the number of parameters and FLOPs of all compared deep learning-based models. Except for DHP, our model has much less computational complexity than other comparative models. Although DHP requires less computation than our model, we can see in Tables 1 and 2 that all the metrics obtained by our model are far superior to those of DHP.

**Table 6: Ablation study for hyperparameters in the loss function.**

| Method | | SAM↓ | ERGAS↓ | PSNR↑ |
|--------|------|------|--------|-------|
| Manual | ×2 | 3.9490 | 3.2144 | 37.6988 |
| Adaptive | | **3.9015** | **3.1983** | **37.7238** |
| Manual | ×4 | 6.0220 | 6.2612 | 31.3210 |
| Adaptive | | **5.6875** | **6.0688** | **31.5756** |
| Manual | ×8 | 8.7427 | 9.9880 | 27.1591 |
| Adaptive | | **8.2716** | **9.8175** | **27.3280** |

**Table 7: Comparison of Params and FLOPs of different methods.**

| Method | Params(M) | FLOPs(G) |
|--------|-----------|----------|
| DHP | 8.58 | 4.65 |
| MCNet | 2.17 | 230.51 |
| ERCSR | 1.59 | 229.3 |
| MSDformer | 32.99 | 24.07 |
| ESSA | 11.52 | 50.19 |
| Ours | 3.81 | 6.37 |

## 3.4 Classification Experiments

To evaluate the effectiveness of the proposed SCPSN in object classification applications, classification experiments were conducted using reconstructed HRHSI images from the Pavia university dataset. We employ the iterative self-organizing data analysis techniques algorithm (ISODATA) to evaluate the results from different SR methods, which is a classical unsupervised semantic segmentation for satellite images. We set the number of the classified category to 5 and the maximum iteration to 10. The visualization results are shown in Figure 8. From the enlarged area, we can observe that the classification result on our result is closest to that on GT, indicating that our method can reconstruct HRHSIs with more accurate texture and spectral information.

## 4 CONCLUSION

This paper proposes a novel network called SCPSN for single-HSI-SR task. In each layer of SCPSN, a SCRB containing SCB, PNAB, and DFB is constructed to reconstruct the HRHSI of the current layer. First, based on the characteristics of HSI, a SCB is designed to utilized the idea of spectral clustering to achieve the filtering of hyperchannels, in order to reduce the computational complexity of the model. Then, considering the non-local similarity of features within the channel, a PNAB is constructed to achieve the reconstruction and enhancement of hyperchannel features. Finally, a DFB is designed to reconstruct all spectral bands in LRHSI by establishing correlations between enhanced hyperchannels and other channels. Extensive experiments conducted on multiple satellite datasets have demonstrated the effectiveness and good generalization ability of the proposed network. In addition, our model has the advantages of fewer parameters and lower computational complexity, and does not need to be trained on large datasets.

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
