# OpenReview forum: "SCPSN: Spectral Clustering-based Pyramid Super-resolution Network  for Hyperspectral Images"
_acmmm.org/ACMMM/2024/Conference — MM2024 Poster_

### Official Review · Reviewer_11Do · 2024-05-11

**Rating:** 4
**Confidence:** 4

**Summary:**

This paper proposes a single hyperspectral image super-resolution method for solving the problem of interference from redundant information in adjacent bands, which leads to spectral and spatial distortion of the reconstruction results as well as an increase in the computational complexity of the model. To address this issue, this paper proposes a spectral clustering-based pyramid super-resolution network (SCPSN) to progressively reconstruct HRHSI at different scales. In each image reconstruction layer, a clustering super-resolution block (CSRB) consisting of spectral clustering block (SCB), patch non-local attention block (PNAB), and dynamic fusion block (DFB) is designed to achieve the reconstruction of detail features.

**Strengths:**

The paper provides a single hyperspectral image super-resolution candidate method, and the experimental results also achieve good performance, but there are still some obvious problems.

**Limitations:**

Here are my main concerns with the paper:
1、The novelty is limited and some of the network structures follow previous work.
2、The main motivation of this paper is to eliminate interference from redundant information in adjacent bands, but the reintroduction of the original LRHS images in the overall network and DFB contradicts this motivation. In addition, there are also doubts about whether there will be information loss in the SCB block.
3、Meanwhile, whether different up-sampling methods such as nearest neighbor interpolation and bilinear interpolation will have an impact on the performance, please add the corresponding experiments.
4、Some of the latest comparison methods can be added and time step T should be explored.

**Suitability:**

2

---

### Official Review · Reviewer_sBoU · 2024-05-16

**Rating:** 3
**Confidence:** 3

**Summary:**

this paper proposes a spectral clustering-based pyramid super-resolution network (SCPSN) to reconstruct HRHSI which can avoid spectral and spatial distortions.

**Strengths:**

The paper propose a network named SCPSN  which can progressively reconstruct HRHSI with rich textures at different scales. Spectral Clustering Block(SCB) is constructed to reduce the computational complexity and number of parameters of the model.

**Limitations:**

1. For Patch Non-local Attention Block (PNAB), the motivation to cut the Q K V into two types of 7 × 7 patches using stride sizes of 4 and 1 is not clear. There is no ablation study for this block.
2. Why not use the CAVE[1] dataset and Harvard[2] dataset which also used in ESSA for experiment?

[1] Fumihito Yasuma, Tomoo Mitsunaga, Daisuke Iso, and Shree K Nayar. Generalized assorted pixel camera: post-capture control of resolution, dynamic range, and spectrum. IEEE transactions on image processing, 19(9):2241–2253,2010.

[2] Ayan Chakrabarti and Todd Zickler. Statistics of real-world hyperspectral images. In CVPR 2011, pages 193–200. IEEE, 2011.

**Suitability:**

2

---

### Official Review · Reviewer_38xL · 2024-05-19

**Rating:** 5
**Confidence:** 4

**Summary:**

This paper proposes a spectral clustering-based pyramid super-resolution network (SCPSN), which utilizes hyperchannels with rich information selected from clustering groups to gradually reconstruct high-resolution hyperspectral images (HR-HSI) at different scales. In each reconstruction layer, a clustering super-resolution block (CSRB) consisting of spectral clustering block (SCB), patch non local attention block (PNAB), and dynamic fusion block (DFB) is designed to achieve the reconstruction of detail features. Extensive experiments validate that the performance of SCPSN is superior to that of some other state-of-the-art (SOTA) HSSR methods in terms of visual effects and quantitative metrics.

**Strengths:**

1.	The paper is generally well written and well-structured. The overall logic of the manuscript is clear, and the language is fluent.
2.	This network has the advantages of short training time, small number of parameters, and low computational complexity.
3.	I think the manuscript has innovative contributions, and sufficient experiments have been conducted to validate their effectiveness. This network does not need to be trained on large-scale datasets.

**Limitations:**

1.	The module boxes of CSRB in Figure 4 are not very clear. Different colors are recommended for better visibility.
2.	The author should clearly state why building datasets instead of directly using public datasets?

**Suitability:**

3

---

### Official Review · Reviewer_qpWc · 2024-05-24

**Rating:** 5
**Confidence:** 3

**Summary:**

This paper proposes a spectral clustering-based pyramid super-resolution network for hyperspectral images. In each pyramid layer of the network, there are mainly three blocks, including spectral clustering block (SCB), patch nonlocal attention block (PNAB), and dynamic fusion block (DFB), which are proposed in this paper, which progressively reconstructs high-resolution HSI.

**Strengths:**

•	A new network structure is proposed with three designed blocks, where the detailed structures of the three blocks are presented in the paper.
•	Three datasets are used to evaluate the proposed method, the results are compared with other methods and show promising results.

**Limitations:**

•	Is the PSNR in the article calculated from just one image? HSI usually uses MPSNR indicator.
•	The experimental results of comparison method ESSA on the Pavia dataset are different from the results in the original paper. what is the reason?
•	Figure 2(b) is a bit misleading, because the neural network block looks the same as Figure 2(a), but the content of the following article is not like this.
•	The datasets used in the experiments are slightly older. It is recommended to consider new datasets, such as Houston-2018.
•	The ablation study did not validate the PNAB part.
•	How many layers does the Pyramid structure have? The ablation study did not verify the impact of the number of layers.

**Suitability:**

3

---

### Meta-Review · Area_Chair_17xj · 2024-07-03

**Recommendation:** Accept (Poster)
**Confidence:** 4

**Metareview:**

The reviews are generally positive towards this work. The authors are suggest to clarify the metrics used to evaluate different algorithms. Particularly, the discrepancy between the experimental results of some comparative methods and their published ones needs to be clarified and explained. Acceptance (poster) is recommended.